



# Evaluating the Impact of Inter-Annual Variability on Long-Term Wind Speed Predictions

Johanna Borowski[1,3], Sandra Schwegmann[1], Kerstin Avila[2,3], and Martin Dörenkämper[1]

[1]Fraunhofer IWES, Fraunhofer-Institut für Windenergiesysteme IWES, Küpkersweg 70, 26129 Oldenburg, Germany
[2]ForWind - Center for Wind Energy Research, Küpkersweg 70, 26129 Oldenburg, Germany
[3]Carl von Ossietzky Universität Oldenburg, School of Mathematics and Science, Institute of Physics, Ammerländer Heerstraße 114-118, 26129 Oldenburg, Germany

**Correspondence:** Johanna Borowski (johanna.borowski@iwes.fraunhofer.de)

**Abstract.** Assessing the wind resource and its associated uncertainties is essential for the profitability of a wind farm, with inter-annual variability in wind speed being a key factor. To estimate the wind resource at a potential wind farm site, a year-long wind measurement campaign is typically conducted and combined with long-term - often numerical - reference data using the Measure-Correlate-Predict (MCP) approach. This process accounts for systematic errors in the reference data and captures the

long-term wind variability of wind speed. Since wind conditions vary from year to year, the selection of a single measurement year within the MCP framework can significantly influence the predicted wind resource. In this study, we systematically evaluate the impact of the measurement year on wind speed predictions using long-term met mast measurements. We also investigate whether classical and advanced machine learning methods can mitigate this sensitivity. Our results reveal that the variation in predicted wind speed due to the chosen measurement year ranges from 1 % to 14 %, depending on the site

and correlation method, with an average of 6.5 %. Excluding years with exceptional wind conditions reduces the mean to 4.2 %. Among the methods selected, the correlation method SpeedSort, along with the advanced machine learning models Random Forest and AdaBoost, most effectively mitigates the influence of inter-annual wind variations in long-term referencing compared to classic linear regression. Additionally, the findings indicate that AdaBoost and Random Forest are especially beneficial for sites with heterogeneous and complex terrain. Furthermore, the study highlights the need for quality-controlled,

long-term datasets across a variety of sites with differing terrain complexities to better understand and manage the effects of inter-annual wind variability in diverse wind climates.

## 1 Introduction

Accurately estimating the wind resource at a potential wind farm site is a critical component of the planning phase and plays a key role in ensuring project profitability. This includes both assessing the wind speeds at the potential site and estimating

associated uncertainties (Rohrig et al., 2019).

To characterize the wind climate at a target site, a wind measurement campaign is usually carried out as a short-term measurement series at least one year as part of the site assessment process (MEASNET, 2022; FGW e.V., 2023).





Since wind conditions vary not only from year to year (inter-annual variability) but also over longer time scales (multi-decadal variability), it is necessary to relate the short-term measurement time series to longer periods (MEASNET, 2022;
FGW e.V., 2023). To achieve this, short-term measurement time series are extended using long-term reference time series which typically span over at least one decade (FGW e.V., 2023).

To extend the one-year short-term measurement with the long-term reference data, a statistical relationship is typically determined based on the overlap time period between the two datasets. The derived correction function is then applied to adjust the long-term data to reflect the wind conditions at the potential site. The resulting hindcast prediction can then be used to
evaluate the wind climate at the target site. This "Measure - Correlate - Predict (MCP)" methodology is widely used in wind resource assessment, with various correlation methods of varying degree of complexity available (Carta et al., 2013).

In industrial wind resource assessments, the MCP methodology is often based on simple linear models (Carta et al., 2013; Basse et al., 2021), but advanced machine learning (ML) applications gain popularity in context of wind energy, (e.g., Optis et al., 2021; Schwegmann et al., 2023; Velázquez et al., 2011; Bass et al., 2000; Zhang et al., 2014; Stetco et al., 2019;
Bakhoday-Paskyabi, 2020; Barber and Nordborg, 2020; Bodini and Optis, 2020; Daniel et al., 2020). Optis et al. (2021) reveal that ML approaches, especially the Random Forest method, enhance the vertical interpolation of wind speeds from surface to hub heights and Schwegmann et al. (2023) conclude that ML applications are beneficial for gap filling with MCP in wind resource assessment.

When estimating the wind resource, it is necessary to account for various uncertainties as they impact the reliability of
assessments. The wind speed uncertainty is a key component and encompasses the uncertainty components such as long-term referencing, MCP methodology, wind measurements, site environment and resolution, wind variability (both inter-annual and inter-monthly, as well as future projections), and other considerations as outlined in the International Electrotechnical Commission (IEC) 61400-15 proposed framework discussed in Lee and Fields (2021). Understanding and mitigating these uncertainties enhances decision-making and risk management in the development and operation of wind energy systems.

Inter-annual wind variability is influenced by various factors - some of which vary significantly by geography - including oceanic and atmospheric oscillations such as El-Niño Southern Oscillation, the North Atlantic Oscillation, and the Pacific Decadal Oscillation. Additionally, the geographical environment, such as mountains or the distribution of land and water masses, shapes wind patterns and can also have an impact on the inter-annual wind variability by altering atmospheric stability or wind direction. These influences can result in significant regional differences in wind climate and its variability.

Numerous methods for determining inter-annual variability can be found in the literature; Lee et al. (2018) compared over 20 of these methods and evaluated their advantages and disadvantages. Depending on the calculation method, datasets study area and the time period considered, the inter-annual variability vary between 1.3 % and 10–15 % (e.g., Lee and Fields, 2021; Wohland et al., 2019; Pullinger et al., 2017; Hamlington et al., 2015; Watson et al., 2015; Früh, 2013; Martin, 2010; Pryor et al., 2006; Klink, 2002; Baker et al., 1990; Justus et al., 1979; Corotis, 1976). The aim of the study is to investigate and mitigate the
impact of inter-annual variability on wind speed predictions in the long-term referencing process of wind resource assessment using MCP methodology.





The first objective is to quantify the impact of inter-annual variability in long-term referencing across multiple sites with varying terrain complexities using classic linear regression as the correlation method in MCP. Furthermore, we examine the sensitivity of both classical and advanced ML methods to inter-annual variability. Based on this analysis, we assess which
methods are most effective in reducing the impact of inter-annual variability in long-term referencing. The study benefits from quality-controlled, long-term measurement time series from twelve wind-energy-relevant sites with diverse terrain and environmental complexities.

The sites, data and the MCP methodology and correlation methods used are introduced in Sect. 2. Key findings related to
site climatology, intercomparison of correlation methods and time range variability are presented in Sect. 3. Finally, Sect. 4 discusses the results, draws conclusions, and provides a brief outlook.

## 2  Data and Methods

This section provides an overview of the site characteristics, the measurement and reanalysis data used in the study. Additionally, it outlines the procedure for defining the impact of inter-annual wind variability in long-term referencing.

### 2.1  Data

### 2.1.1  Tall met mast data

To investigate the inter-annual variability of the wind climate, long-term continuous wind measurements are required. Numerous wind measurements exist at a height of 10 m (Ramon et al., 2020); but vertically interpolating these to wind turbine hub heights can introduce additional uncertainties. Therefore, this study focuses on measurement data from tall met masts that provide wind measurements at around 100 m - close to the hub heights of modern wind turbines. Additionally, the selected

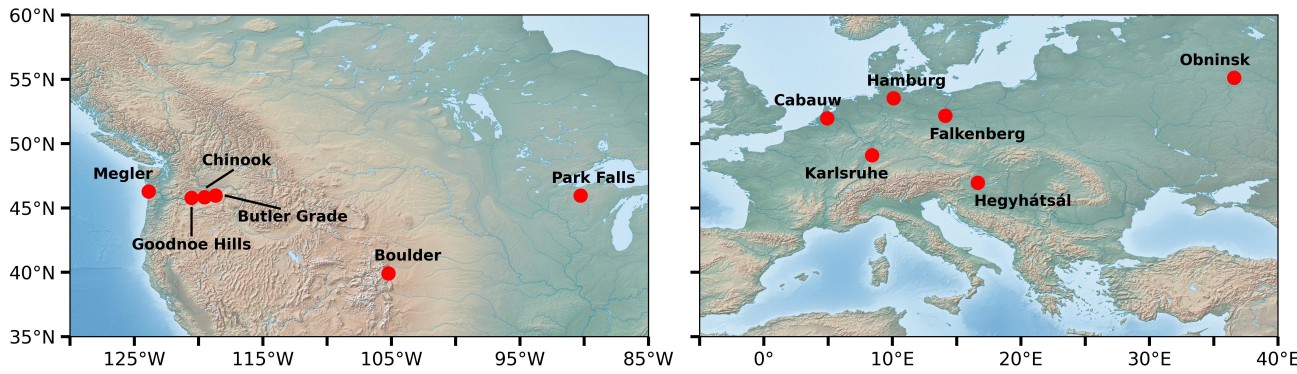

**Figure 1.** Distribution of the met mast positions from the Tall Tower Dataset (Ramon et al., 2020), including additional masts at the sites Cabauw, Hamburg, Falkenberg, and Karlsruhe.






sites have a yearly average wind speed of more than $4\,\mathrm{m\,s^{-1}}$ at wind energy relevant heights and provide long-term wind measurements – each more than seven consecutive overlapping years (2010–2016).

A total of eight met mast locations from the Tall Tower Dataset (Ramon et al., 2020) were identified as suitable for the study (Table 1): Megler (US), Goodnoe Hills (US), Chinook (US), Butler Grade (US), Boulder (NWTC M2 (Jager and Andreas, 1996), US), Park Falls (WLEF (Davis et al., 2003), US), Hegyhátsál (HU) and Obninsk (RU). Additionally, data from four further met masts were considered with the sites: Hamburg (Hamburg Weather Mast, GER), Falkenberg (GER), Karlsruhe (Kohler et al. (2018), GER), and Cabauw (NL).

**Table 1.** Overview of the met mast positions and measurement parameters. NWTC: National Wind Technology Center; WS: wind speed; WD: wind direction.

| Site | Country | Lon | Lat | Measurement height [m] | Mean wind speed [m s$^{-1}$] | Measuring devices | Resolution of available data |
|---|---|---|---|---|---|---|---|
| Megler | US | -123.88 | 46.27 | 53 | 5.22 | unknown | 10 min |
| Goodnoe Hills | US | -120.55 | 45.78 | 59 | 5.97 | unknown | 10 min |
| Chinook | US | -119.53 | 45.83 | 50 | 4.64 | unknown | 10 min |
| Butler Grade | US | -118.68 | 45.95 | 62 | 7.23 | unknown | 10 min |
| Boulder (NWTC M2) | US | -105.23 | 39.91 | 80 | 4.72 | cup/wind vane | 10 min |
| Park Falls (WLEF) | US | -90.27 | 45.95 | 122 | 6.07 | unknown | hourly |
| Cabauw | NL | 4.93 | 51.97 | 80 | 6.77 | cup/wind vane | 10 min |
| Karlsruhe | GER | 8.43 | 49.09 | 100 | 4.67 | cup/wind vane | 10 min |
| Hamburg (Hamburg Weather Mast) | GER | 10.10 | 53.52 | 110 | 6.36 | sonic | 10 min |
| Falkenberg | GER | 14.12 | 52.17 | 98 | 5.91 | sonic | 10 min |
| Obninsk | RU | 36.6 | 55.11 | 121 | 5.48 | unknown | hourly |
| Hegyhátsál | HU | 16.65 | 46.96 | 82 (WS), 115 (WD) | 4.84 | sonic | hourly |

The selected met masts are distributed across the Northern Hemisphere (Fig. 1), primarily concentrated in Central Europe and the northern states of the United States. They are located in various types of terrain complexities. To classify the complexity, the standard deviation of terrain height ($\sigma_{terrain}$) within a 10 km x 10 km grid surrounding the met masts was calculated. The following categories of terrain complexity were established: simple ($\sigma_{terrain} < 10\,\mathrm{m}$), heterogeneous ($10\,\mathrm{m} \leq \sigma_{terrain} < 20\,\mathrm{m}$), complex ($20\,\mathrm{m} \leq \sigma_{terrain} < 100\,\mathrm{m}$) and very complex ($\sigma_{terrain} \geq 100\,\mathrm{m}$). Based on this classification, the sites were assigned to the appropriate categories according to their terrain complexity (Table 2).





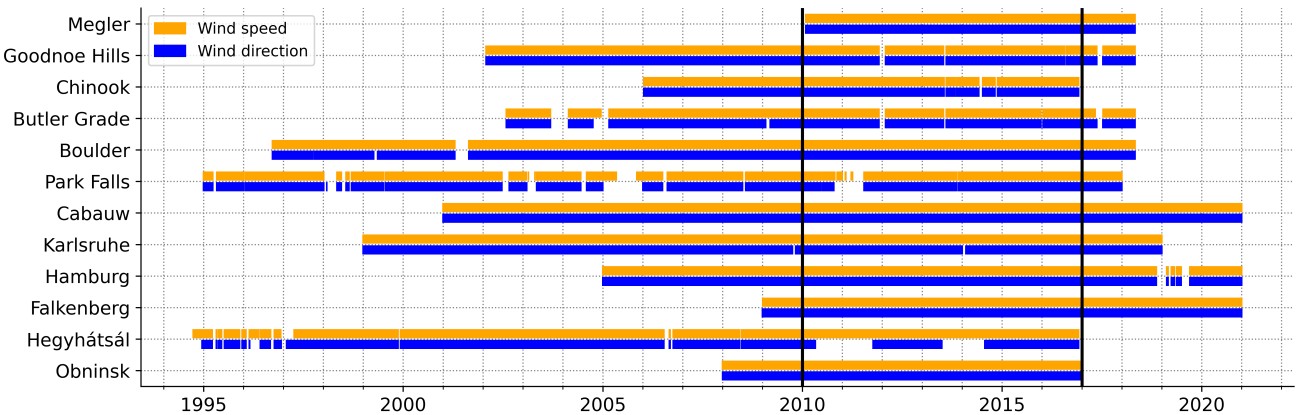

**Figure 2.** Measured wind speed and direction availability at the selected sites from end of 1994 to 2021, with an overlapping period from 2010 to 2016 highlighted by black lines. Within this overlapping period the individual sites exhibit at least 70 % wind data availability per year, except for Park Falls, which falls below 50 % in 2010 and 2011 and Hegyhátsál with less than 50 % wind direction data in 2010, 2011, 2013, and 2014.

Furthermore, the measurement data for this study are available at various averaging intervals (Table 1). Since the reanalysis
data are provided at hourly resolution, the measurements are aggregated to hourly intervals unless they are already available in hourly format.

### 2.1.2 Model data

Meteorological reanalysis data provide long-term datasets of relevant meteorological quantities by combining historical observation data with advanced numerical weather models. Modern reanalysis datasets provide data over periods of more than 50
years thus enable to analyze wind variability. In wind resource assessment, reanalysis data from global circulation models are used as long-term reference datasets to provide a basis for the inter-annual and decadal variability of the wind conditions. In this study, the reanalysis data ERA5 (Fifth Generation European Reanalysis, Hersbach et al. (2020)) of the European Centre for Medium-Range Weather Forecasts (ECMWF) are used. Due to its high time resolution of one hour and spatial resolution of ~0.28°(31 km) and an availability of the wind speed components at 100 m, the reanalysis has been proven to be well suited for
several wind energy applications (Olauson, 2018; Hahmann et al., 2020; Dörenkämper et al., 2020; Gottschall and Dörenkämper, 2021). The ERA5 reanalysis data cover the time period from 1940 till present and thus, it gives the possibility to capture the decadal changes in the atmosphere within this period.

In order to compare ERA5 with measurements and to use the data for further analysis, ERA5 data have been extracted from the nearest grid point to the site of interest, i.e. the met mast locations described above. The analysis covers the time range
from 1950 to 2020.





**Table 2.** Description of site characteristics and terrain complexity based on the standard deviation ($\sigma_{terrain}$) [m] of terrain height variation within a 10 km x 10 km area around each site.

| Site | Landscape | $\sigma_{terrain}$ [m] | Terrain complexity category |
|------|-----------|------------------------|------------------------------|
| Cabauw | coastal area | 1.8 | simple |
| Park Falls | boreal lowland, wetland forests, flat | 9.8 | simple |
| Hamburg | urban, flat | 14.7 | heterogeneous |
| Falkenberg | forested, fields, small lakes, flat | 17.7 | heterogeneous |
| Obninsk | forested, fields, urban, flat | 19.9 | heterogeneous |
| Karlsruhe | forested, Rhine plain | 25.0 | complex |
| Hegyhátsál | rural, fields, forest patches, flat | 25.6 | complex |
| Chinook | near river/valley, fields | 43.2 | complex |
| Megler | North: forested and hilly, East to South: hilly with a river and estuary, West: coastal | 86.2 | complex |
| Butler Grade | near valley, mountainous | 177.6 | very complex |
| Goodnoe Hills | near valley, mountainous | 195.1 | very complex |
| Boulder | West: Rocky Mountains, East: flat/open | 266.9 | very complex |

## 2.2 Methods

### 2.2.1 Measure-Correlate-Predict (MCP)

In the wind energy industry, the state-of-the-art procedure for correcting reference data to site-specific conditions (mostly based on measurement data) is known as the **M**easure-**C**orrelate-**P**redict (MCP) method. In MCP, on-site measurements are
correlated with a long-term (typically numerical) reference dataset, and predictions are then made based on this correlation, accounting for the systematic errors that long-term datasets often exhibit. In this study, several MCP methods are used and compared with each other, and are introduced in the following.

**Classical approaches**

In the classical approaches, the correlation between the short- and long-term dataset is done based on the full overlapping
period of both datasets. Three correlation methods using linear regression are described in the following and are used in this study:

– Classic linear regression (Clas-LinReg): It is commonly used in wind resource assessment to calculate the correlation in the MCP methodology (Carta et al., 2013).



- Sector-wise linear regression (Sec-LinReg): Modification of the classic linear regression method, which considers a division of the wind data into individual wind direction sectors. In this study, a binning in $30°$ sectors is applied.

- SpeedSort (King and Hurley, 2005): Wind speed values are sorted in ascending order step-wise before performing classic linear regression. The size of the steps are set to $1\,\mathrm{m\,s^{-1}}$ in this study.

**Advanced approaches**

The advanced approaches in this study for predicting wind speed conditions at a specific site use ML models as regression tools within the MCP framework. The input data are divided into training, testing, and validation datasets. In this case, the data from the overlapping time period are randomly selected within a year to create a split of $70\,\%$ training and $30\,\%$ test data. Although this reduces the amount of sample size, it still captures all relevant characteristics - such as daily and seasonal cycles - due to the randomness of the selection process. The choice of sub-sample regression models is based on the findings of Schwegmann et al. (2023) using the Python package *scikit-learn (sklearn)* (Pedregosa et al., 2011), and includes:

- ML linear regression (ML-LinReg): Similar to classic linear regression, but the training data are randomly selected from the overlapping time period. No additional features are included.

- Random Forest (RForest) Regressor (Grömping, 2009): An ensemble-based regression method that combines multiple uncorrelated decision trees. Each tree has a limited depth and is trained on a different subset of the training dataset. The final prediction is obtained by averaging the outputs of all individual trees, each given equal weight.

- AdaBoost Regressor (Solomatine and Shrestha, 2004; Freund and Schapire, 1997; Drucker, 1997): The AdaBoost Regression is an ensemble-based method that uses decision trees. The main difference to the RForest method is that each tree has only one node and two leaves. Additionally, individual weights are applied to combine the results from each tree into a single final prediction. Data points whose prediction appears more complex than others are weighted stronger and the next tree grows on the weighted previous tree, taking its errors into account. The goal of this method is to combine multiple weak learners into a more accurate and robust model.

- K-Nearest-Neighbours (KNN) Regressor (Fukunaga and Narendra, 1975): KNN is a simple ML approach, which is based on the average of the k-nearest neighbors of all features in the reference dataset, weighted by their distance of similarity.

In order to find the best combination of defined model parameters - such as the number (k) of nearest neighbors (KNN) or the optimal tree depth (RForest), we use the GridSearchCV function provided by the *scikit-sklearn* package (LaValle et al., 2004) for model training. The negative mean squared error is used as the evaluation score. The parameter combination with the lowest score is then used for performing the predictions.

As wind speed related and atmospheric state descriptive features for the advanced ML approaches, wind direction, temperature, pressure, time of day and seasonality of the ERA5 dataset are used partly decomposed into sine and cosine to account for the circular nature of the quantity (Table 3).




**Table 3.** Input features derived from the reference dataset for the KNN, Random Forest, and AdaBoost regression models.

| Input feature |
| --- |
| 100 m wind speed |
| Sine and cosine of 100 m wind direction |
| 2 m air temperature |
| Mean sea level pressure |
| Sine and cosine of time of day |
| Sine and cosine of month |

### 2.2.2 Hindcast ensemble range

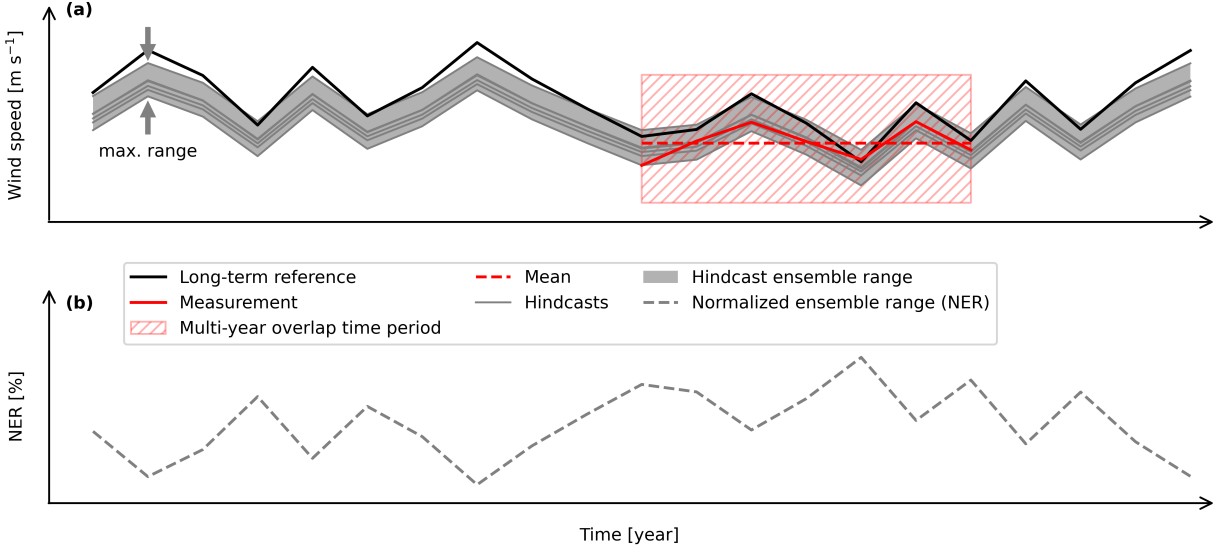

**Figure 3.** Schematic illustration of the max. hindcast ensemble range: (a) Annual mean of measurement (red) and long-term reference (black) data with multi-year overlap time period (red hatched). Annual mean wind speed predictions (hindcasts, gray lines) based on individual years of the multi-year overlap period, along with the hindcast ensemble range (gray shaded area). (b) By the measured mean wind speed data (red dashed) normalized hindcast ensemble range (NER [%], gray dashed line).

The impact of inter-annual variability on the wind resource is analyzed by considering the maximum ensemble range of wind speed predictions. This is done as follows (Fig. 3): In a first step, for each individual year within the multi-year overlap period (red hatched) a wind speed prediction with MCP is performed. The resulting wind speed predictions (hindcasts; Fig. 3a, gray





lines) spanning the time range of the long-term reference data, vary due to year-to-year wind variability, with higher variability at a site leading to a wider spread.

To fully capture the impact of inter-annual wind variability on the hindcast, the maximum range of the hindcast ensemble is analyzed (Fig. 3a). This maximum hindcast ensemble range (ER) is defined as the difference between the maximum and minimum wind speeds in the ensemble for each year. The time average of these over all years is referred to as MER (**m**ean

max. **e**nsemble **r**ange).

To ensure comparability between sites, the maximum **e**nsemble **r**ange is **n**ormalized (NER [%], Fig. 3b) by the mean measured wind speed of the multi-year overlap period (Fig. 3a, red dashed). Accordingly, the **m**ean **n**ormalized max. **e**nsemble **r**ange (MNER [%]) is the time average of the NER values. A high (low) MER, NER, or MNER value indicates a broader (narrower) maximum ensemble range, which corresponds to greater (lower) inter-annual variability at the site. This entire pro-

cedure is repeated for each site and each correlation method used in the MCP approach. The MCP itself is based on hourly ERA5 data and measured wind speed data.

## 3 Results

### 3.1 Wind climatology of investigated sites

The selected sites exhibit varying wind climates, which are outlined below. In addition, the differences between the reference

time series (ERA5) and the measurements are examined. The wind characteristics of the measurement data and the ERA5 reanalysis data for the multi-year overlap time period of 2010–2016 for all sites used in this study are illustrated in Figure 4 and Figure 5. At most sites, the measured wind direction is predominantly westerly (Fig. 4) with varying intensity, except in Hegyhátsál and Boulder, where northern winds are also more frequent. Sites with simpler terrain exhibit a more uniform wind direction distribution (e.g. Cabauw) compared to those with complex terrain (e.g. Butler Grade). The measured wind speed for

most sites (e.g. Cabauw, Falkenberg, Boulder) follows a Weibull distribution, while some (e.g. Goodnoe Hills) display bimodal distributions, which are common in highly complex terrain due to redirection of the larger-scale flow.

When comparing reanalysis data with the measurements, the main wind direction distribution is generally captured by the ERA5 data. But, in some cases, deviations occur between wind direction sectors, with wind directions shifted to an adjacent sector (e.g., Karlsruhe, Chinook, Butler Grade). At the Megler site, an underrepresentation (overrepresentation) of the easterly

(northwesterly) winds can be observed in the ERA5 data. This could be attributed to the highly variable wind direction sectors that are influenced by terrain characteristics (Table 2). At the Boulder site, the ERA5 data indicate a shift of nearly 90° compared to the measured wind direction. This can be attributed to the sites's location at the foot of the Rocky Mountains, where local geographical features strongly influence wind patterns. Furthermore, the ERA5 reanalysis data overrepresent lower wind speeds and underrepresent higher wind speeds at certain sites (Fig. 5). At the three most complex sites, lower wind

speeds are overrepresented, while the highest wind speeds are almost completely absent in the ERA5 data. Conversely, at sites Hegyhátsál, Chinook and Megler, lower wind speeds are slightly underrepresented, while higher wind speeds occur more frequently in ERA5 than in the measurements. The differences between ERA5 and measurement data may arise from choosing



**Figure 4.** Wind direction distribution of measurement (blue) and ERA5 reanalysis (orange) data for the time period 2010–2016. Note: Overlapping sectors lead to a brownish color. Sites are sorted by terrain complexity from simple to very complex from top left to bottom right.

the nearest ERA5 grid point to the mast location, which may not accurately reflect the site characteristics due to ERA5's resolution.



**Figure 5.** Wind speed distribution of measurement (blue) and ERA5 reanalysis (orange) data for the time period 2010–2016. Note: Overlapping bars lead to a brownish color. Sites are sorted by terrain complexity from simple to very complex from top left to bottom right. A Weibull fit (solid lines) is performed and shape ($\lambda$ [m s$^{-1}$]) and scale ($k$) parameters are provided in the legend except for cases with distinct non-Weibull, e.g. bimodally distributed met mast winds.



**Figure 6.** Maximum wind speed hindcast ensemble range (shaded) and individual hindcasts (solid lines) for sites in Europe **(a)** and the United States **(b)** are based on individual years from 2010 to 2016 using classical linear regression in the MCP method. **(c)** The normalized max. hindcast ensemble range (NER [%]) is normalized with the averaged measured wind speed from 2010 to 2016.

## 3.2 Wind speed hindcast based on classic linear regression

To assess the impact of inter-annual wind variability on long-term referencing, a wind speed hindcast ensemble derived from individual years spanning 2010 to 2016 is analyzed using classical linear regression as the correlation method in the MCP process. The hindcast ensemble is illustrated in Figure 6 for several sites in Europe (a) and in the US (b), respectively.

Those sites are affected by different weather regimes due to their geographical locations, which shape the inter-annual variations in the wind climate. Central European sites follow a similar pattern, while eastern sites like Obninsk and Hegyhátsál





**Table 4.** Overview of the mean max. hindcast ensemble range (MER [m s$^{-1}$]) and by the mean measured wind speed normalized max. hindcast ensemble range (MNER [%]) for the time period from 2010 to 2016 and 2012 to 2016.

| Site | MER [m s$^{-1}$] | | | MNER [%] | | |
| --- | --- | --- | --- | --- | --- | --- |
|  | 2010–2016 | 2012–2016 | Difference | 2010–2016 | 2012–2016 | Difference |
| Cabauw | 0.14 | 0.1 | 0.04 | 2.1 | 1.5 | 0.6 |
| Park Falls | 0.26 | 0.11 | 0.15 | 4.2 | 1.8 | 2.4 |
| Hamburg | 0.44 | 0.41 | 0.03 | 7.0 | 6.6 | 0.4 |
| Falkenberg | 0.05 | 0.04 | 0.01 | 0.9 | 0.7 | 0.2 |
| Obninsk | 0.6 | 0.23 | 0.37 | 11.0 | 4.3 | 6.7 |
| Karlsruhe | 0.2 | 0.15 | 0.05 | 4.3 | 3.3 | 1.0 |
| Hegyhátsál | 0.53 | 0.34 | 0.19 | 11.0 | 7.0 | 4.0 |
| Chinook | 0.29 | 0.29 | 0.00 | 6.3 | 6.4 | -0.1 |
| Megler | 0.74 | 0.38 | 0.36 | 14.1 | 7.2 | 6.9 |
| Butler Grade | 0.33 | 0.32 | 0.01 | 4.6 | 4.5 | 0.1 |
| Goodnoe Hills | 0.22 | 0.19 | 0.03 | 3.7 | 3.3 | 0.4 |
| Boulder | 0.51 | 0.26 | 0.25 | 10.8 | 5.5 | 5.3 |

diverge due different i.e. more continental climate. Similarly, in the US, Butler Grade, Goodnoe Hills, and Chinook located west of the continental divide share a common pattern that is different to sites further east.

The dispersion of the individual hindcasts within the ensemble range varies depending on the site (Fig. 6a and b, shaded areas). The density distribution of the individual hindcasts within the ensemble is displayed to the right of the corresponding time series for each site (Fig. 6). At the sites Falkenberg and Cabauw, there are only small differences between the individual hindcasts, resulting in a narrow density distribution (Fig. 6a). However, at most remaining sites broader density distributions are found. While, e.g., at site Chinook the broad density distribution is caused by evenly distributed hindcasts. At other sites such as Megler and Boulder in the US as well as Hegyhátsál and Obninsk in Europe, individual outliers dominate the ensemble spread. In most cases these outliers are related to the years 2010 and 2011 in the MCP process (see below). Furthermore, some sites, e.g. Butler Grade and Hamburg have a bimodal density distribution with individual hindcasts clustering at the upper and lower limits of the ensemble. At the site Hamburg, the first (last) years of the multi-year overlap period are related to the upper (lower) hindcasts, which could be connected to a decreasing trend in the measurements for the period 2010 to 2016 (not shown).

Considering the ensemble range of the sites, the mean max. hindcast ensemble range (MER [m s$^{-1}$]) over the period 1950 to 2020 is analyzed and varies between 0.1 and 0.7 m s$^{-1}$ depending on the site (Table 4, MER). To enhance the comparison of the sites, the normalized hindcast ensembles range (NER [%]) for each site from 1950 to 2020 is illustrated in Figure 6c





and the corresponding time averaged values are presented in Table 4 (MNER [%]). The MNER has values between 0.9 %
to 14 % with a certain clustering of sites (Fig. 6c). The lowest values of less than 2 % can be found at the sites Cabauw
(simple terrain) and Falkenberg (heterogeneous terrain). Most of the sites exhibit ranges between 3 % and 7 % (simple to
(very) complex), grouped into a cluster from 3 % to 5 % and another from 6 % to 7 %. Three of the remaining sites (Boulder
(very complex terrain), Hegyhátsál (complex terrain) and Obninsk (heterogeneous terrain) cluster around 11 %, while Megler
(complex terrain) exhibits the highest value of approx. 14 %. Regarding a potential connection between terrain complexity

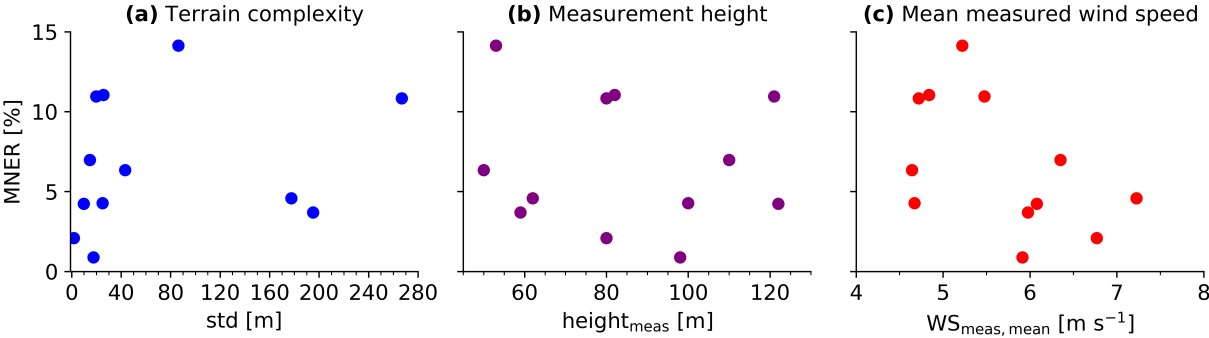

**Figure 7.** Correlation between MNER [%] (mean max. hindcast ensemble range) and terrain complexity **(a)** as the standard deviation (std
[m]) for a 10 km x 10 km grid around the met mast, measurement height **(b)** and mean measurement wind speed **(c)**, respectively, based on
the overlapping time period 2010 to 2016.

and MNER, it appears that a clear statistical relationship remains unclear (Fig. 7a). Furthermore, there is no indication of a
correlation between the MNER and the used measurement height (Fig. 7b) or the measured average wind speed (Fig. 7c).
Excluding the years 2010 and 2011, which were mentioned above as being responsible for a larger ensemble range at certain
sites, the MER reduces to 0.1 to 0.4 m s$^{-1}$ and the MNER reduces to values from 0.7 % to 7.2 %, whereas the four sites Megler,
Boulder, Hegyhátsál, Obninsk decrease most to values between 4 and 7 % (Table 4). Especially at the site Megler, the two years
almost doubled the MNER [%].
The site's wind climate and the deviation of an individual year from the wind climate of a site emerge as significant factors on
the wind speed hindcast based on classic linear regression. In the following sections, further classical and complex regression
methods are analyzed whether they can reduce the influence of inter-annual variability in the long-term referencing process.

### 3.3   Intercomparison of MCP approaches

In addition to classical linear regression, other regression models in MCP are used for long-term referencing. Moreover, ad-
vanced machine learning (ML) methods become increasingly attractive as correction methods in MCP. This section expands
the previous analysis to include six additional regression models, including advanced ML methods. The aim is to evaluate the
performance of the various methods and assess whether any of these methods, particularly the advanced ML methods, have





the potential to reduce uncertainty in estimating inter-annual variability. Furthermore, the analysis compares the methods with respect to their sensitivity across different wind climatologies.

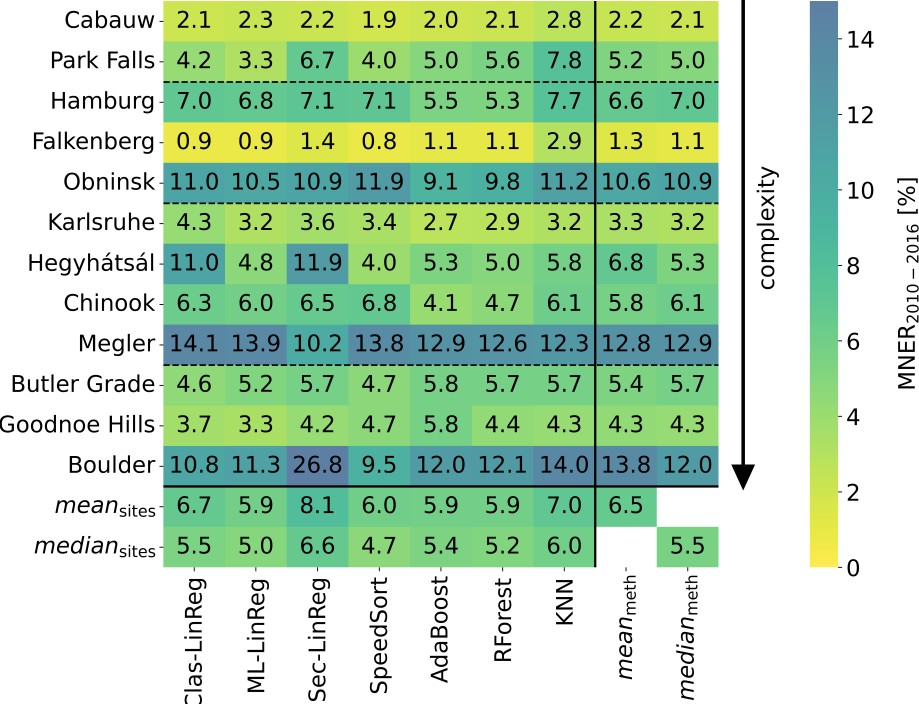

**Figure 8.** Comparison of the mean max. normalized ensemble range based on the multi-year overlap time period 2010–2016 ($\text{MNER}_{2010-2016}$ [%], illustrated by the color scale) as obtained by different methods (indicated below) and for varying sites (indicated on the left). The sites are sorted by the terrain complexity from lowest to highest (black arrow); terrain complexity categories (simple, heterogeneous, (very) complex) are separated by dashed horizontal black lines.

The MNER based on the years 2010 to 2016 is illustrated for all sites and correlation methods in Figure 8, where the Clas-LinReg column matches that in Table 4. Across all sites and the six additional methods, the MNER varies from less than 1 % to approximately 14 %, with an outlier at Boulder with 27 % for sector-wise linear regression. Aside from this outlier, the same MNER is observed as for classical linear regression of the previous section. Averaged across all sites and correlation methods (including Clas-LinReg), the MNER is 6.5 % (median: 5.5 %).

Regardless of the correlation approach, the MNER tends to be higher at certain sites, with no clear dependence on terrain complexity (Fig. 8): The lowest $mean_{\text{meth}}$ (mean across all methods) is found for the sites Falkenberg (heterogenous terrain), Cabauw (simple terrain) and Karlsruhe (complex terrain) with 1 % to 3 %. Most of the other sites reveal a MNER from approx. 4 % up to 7 % including all complexity types (except simple) and measurement heights. The highest $mean_{\text{meth}}$ values are found for Obninsk (heterogeneous terrain), Boulder (very complex terrain) and Megler (complex terrain) with MNERs of approx.

11 % to 14 %, what is similar to the results from the previous section (Sect. 3.2) for the Clas-LinReg except for Hegyhátsál.





Inter-comparing the seven methods, the simple methods ML-LinReg and SpeedSort as well as the advance models RForest and the AdaBoost exhibit a MNER averaged across all sites ($mean_{\text{sites}}$) of about 6 %. But, considering the $median_{\text{sites}}$, the SpeedSort method exhibits the lowest value of 4.7 %, followed by ML-LinReg (5 %), RForest (5.2 %) and AdaBoost (5.4 %). The Clas-LinReg as well as KNN and the Sec-LinReg are most affected by the inter-annual variability, resulting in a $mean_{\text{sites}}$

MNER between 6.7 % to 8 %.

More specifically, *KNN* indicates the highest MNER across multiple sites. While the KNN method is fast and simple, there is no indication that it effectively reduces the impact of inter-annual wind variability in hindcast wind speed predictions at any site.

The *Sec-LinReg* indicates varying affects on inter-annual variability, depending on the site, data quality and availability. There

are indications that the high values in the MNER result from a limited sample size of wind data for specific wind directions related to site-specific climate conditions or general data loss due to measurement failures rather than inter-annual variability. The Megler and Boulder sites exhibit significantly different terrain characteristics for varying wind directions (Table 2). But, while the MNER decreases for Megler, the Boulder site, including the wind direction, results in a MNER of 26.8 %. Details reveal an insufficient sample size at Boulder to establish a representative correlation function within the MCP process for certain

sectors (not shown). There is strong evidence that this issue is linked to the nearly 90-degree shift between the measurements and ERA5 data (Fig. 4), arising from the complex terrain that is not accurately represented by the nearby ERA5 data point used. At the sites Park Falls and Hegyhátsál, there is a loss of wind data (Fig. 2) due to measurement failures in 2010/11 and 2013/14 (Hegyhátsál only), which adversely impacted the accuracy of wind speed predictions.

The *SpeedSort* has a normalized ensemble range averaged over all sites, which is about 0.7 percentage points lower than that

of Clas-LinReg. At the sites such as Hegyhátsál, Park Falls and Boulder, where the ensemble range for Sec-LinReg is higher, SpeedSort provides lower values and could be a suitable method in such cases.

The advanced ML approaches *RForest and AdaBoost* have the potential to reduce the MNER compared to the Clas-LinReg method, with the reduction varying by site and depending on terrain complexity. The reductions occur in heterogeneous (except Falkenberg) and complex terrain, while no reduction is noted in simple and very complex terrain. This could be due to the

limited number of sites; a larger number of sites per complexity category would be beneficial to gain further insights.

Comparing the *Clas-LinReg* with the *ML-LinReg* method, the normalized ensemble range results in almost all sites in similar values. Only at some sites, the values for the ML linear regression are slightly reduced (up to 1 %), while at the Hegyhátsál site, a reduction of 6 % is observed.

Overall, the impact of inter-annual wind variability vary between 1 % and 14 %, depending on the site and method. Advanced

ML methods have the potential to reduce the MNER, but the magnitude depends on the site. The impact of inter-annual variability average across all sites and methods is 6.5 % (median 5.5 %).





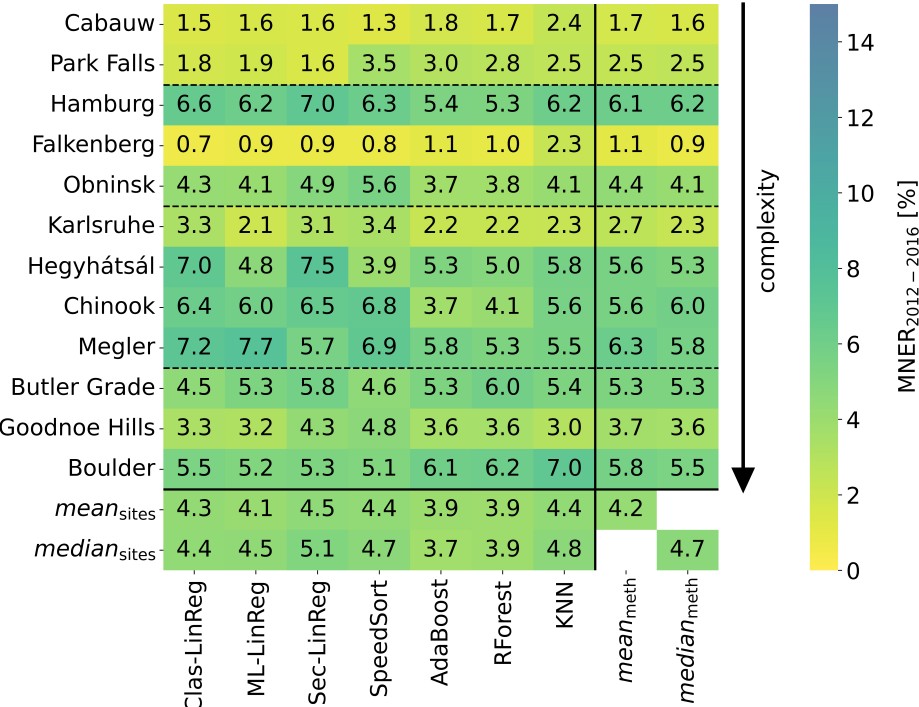

**Figure 9.** Comparison of the mean max. normalized ensemble range based on the multi-year overlap time period 2010–2016 (MNER$_{2012-2016}$ [%], illustrated by the color scale) as obtained by different methods (indicated below) and for varying sites (indicated on the left). The sites are sorted by the terrain complexity from lowest to highest (black arrow); terrain complexity categories (simple, heterogeneous, (very) complex) are separated by dashed horizontal black lines.

## 3.4 Variation of multi-year overlap time period

Varying the multi-year overlap period is crucial for assessing result robustness across different time frames. Thus, the overlap period will be varied in the following subsections. First, it will be shortened to analyze the impact of years with deviating wind climate (Sect. 3.4.1), and second, it will be extended to examine result consistency over time (Sect. 3.4.2).

### 3.4.1 Impact of years with a deviating wind climate

In Sect. 3.1, it was observed that the hindcasts based on the measurement years 2010 and 2011 deviate at many sites. At the beginning of 2010, a strong El Niño event took place, followed by a La Niña event towards the end of 2010 and throughout 2011 (NOAA/National Weather Service, 2025) which can affect wind speed and has a high impact on inter-annual wind variability (e.g., Mohammadi and Goudarzi, 2018; Li et al., 2010). Thus, in the further analysis both years, 2010 and 2011, are excluded, and the impact of their exclusion on long-term referencing is evaluated.





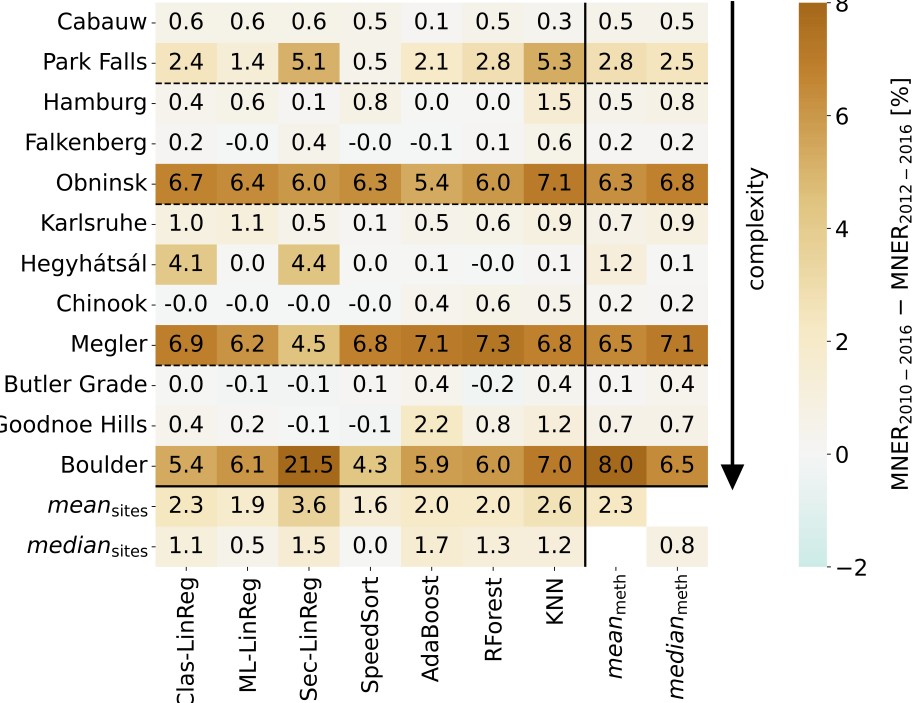

**Figure 10.** Comparison of the difference of the mean max. normalized ensemble range based on the multi-year overlap time period 2010–2016 ($MNER_{2010-2016}$ [%]) and 2010–2016 ($MNER_{2010-2016}$ [%], illustrated by the color scale) as obtained by different methods (indicated below) and for varying sites (indicated on the left). The sites are sorted by the terrain complexity from lowest to highest (black arrow); terrain complexity categories (simple, heterogeneous, (very) complex) are separated by dashed horizontal black lines.

After the exclusion of 2010/11, the MNER lies between approx. 1 % to 7 % depending on the site and method (Fig. 9). Averaged across all methods, the lowest $mean_{meth}$ values of less than 3 % are related to the simple and heterogeneous sites Cabauw, Falkenberg and Park Falls, but also to the complex site Karlsruhe. The sites Obninsk (heterogeneous terrain) and Goodnoe Hills (very complex terrain) follow with $mean_{meth}$ values below 5 %, while the remaining sites have $mean_{meth}$ MNER between 5 % and 6 %.

Averaged across all sites and methods, the MNER is 4.2 % (Fig. 9) representing a decrease of 2.3 %-points (Fig. 10) compared to the time period from 2010 to 2016 (Fig. 8). This can be especially attributed to the decrease of the MNER at sites with high to very high MNER for the time period 2010 to 2016 (Obninsk, Megler, Boulder). Averaged across all methods, the MNER decreases between 2.9 % (Park Falls) and 8 % (Boulder) (Fig. 10). The highest decrease is observed for the Sec-LinReg method at the site Boulder with a reduction of 21.5 %.

Furthermore, excluding the years 2010/11 at the Hegyhátsál site significantly reduces the ensemble range, particularly for the Clas-LinReg and Sec-LinReg methods. For clarification, the deviations between the difference values in Table 4 and Figure





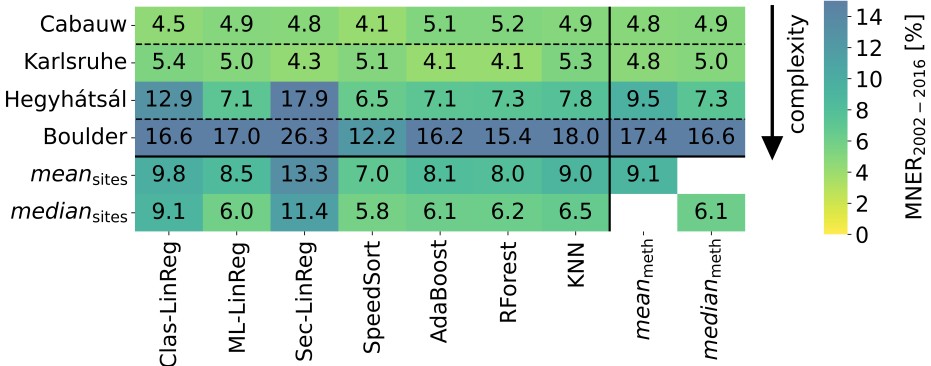

**Figure 11.** Comparison of the mean max. normalized ensemble range based on the multi-year overlap time period 2002–2016 ($MNER_{2002-2016}$ [%], illustrated by the color scale) as obtained by different methods (indicated below) and for varying sites (indicated on the left). The sites are sorted by the terrain complexity from lowest to highest (black arrow); terrain complexity categories (simple, complex, very complex) are separated by dashed horizontal black lines.

10 for the Clas-LinReg are related to rounding errors. Additionally, the previously noted difference between ML-LinReg and
Clas-LinReg persists, with ML-LinReg demonstrating an approximately two percentage points lower MNER (4.8 % vs. 7.0 %).

The RForest and AdaBoost can reduce the impact of inter-annual variability on long-term prediction, especially at sites with heterogeneous (e.g., Hamburg) and complex (e.g., Chinook) terrain complexity (Fig. 9). At simple and very complex sites, ML-LinReg or classical approaches could be suitable options.

Averaged across all sites ($mean_{sites}$), there are only slight deviations between the MNER for the various correlation methods
(0.6 %-points) compared to the time period 2010 to 2016 (2 %-points). The lowest $mean_{sites}$ are achieved by the advanced ML methods AdaBoost and RForest (3.9 %), followed by the ML-linear regression with 4.1 %. The Sec-LinReg reduces most with approx. 3.6 %-points excluding 2010/11, but is still the method with the highest value (4.5 %).

In general, it can be concluded that the impact of individual years with a differing wind climate is higher than the choice of the correlation method within the MCP approach. However, by selecting the correlation method in long-term referencing, the
impact of inter-annual variability can be reduced, but the magnitude depends on the site.

### 3.4.2 Long-term multi-year overlap

In the previous sections, time periods of five (Sect. 3.2) and seven (Sect. 3.4.1) years were analyzed. This section extends the multi-year overlap time period to 15 consecutive years from 2002 to 2016 which allows analyzing the robustness of the ensemble range due to inter-annual variability. But, extending the multi-year period by such extent limits the analysis to four
sites, as not all provide data for such a long time period (Fig. 2). In Figure 11 the MNER is presented for the four remaining sites.



Depending on the site and method, the MNER for the 15 year period results in values from 4 % up to 18 % (except for Sec-LinReg at Boulder). Comparing the MNER based on the seven and 15 years multi-year overlap periods, the MNER more than doubles at the simple terrain site Cabauw. At the complex site Karlsruhe, the MNER averaged over all methods increases

by approx. 1.5 %-points. For the remaining sites Hegyhátsál (heterogeneous terrain) and Boulder (very complex terrain), the increase of the $mean_{\mathrm{meth}}$ is approx. 2 % and 4 % points, respectively. This increase in the MNER for the 15-year overlap, indicating that the full inter-annual variability is not captured by the 7-year period and that the increase is likely linked to further years with deviating wind climate within the extended time period.

Comparing the methods at the Cabauw site, the SpeedSort has the lowest MNER at 4.1 % (Fig. 11). For the complex

Karlsruhe site, the advanced ML methods AdaBoost and RForest have the lowest value, followed by the Sec-LinReg. At the remaining and (very) complex sites Hegyhátsál and Boulder, the SpeedSort method followed by the advance ML methods Random Forest and AdaBoost have the lowest MNER. In conclusion, the SpeedSort, Random Forest and AdaBoost methods have the lowest MNER for the extended 15 years multi-year overlap time period.

Bringing together the findings from the previous analysis, the results offer valuable insights into the inter-annual variability related sensitivities of classical and advanced ML methods in the long-term referencing (MCP) process. They illustrate that selecting the appropriate method can substantially reduce the uncertainties tied to inter-annual variability, where advanced ML methods can make a contribution. Moreover, the analysis highlights the crucial need for thorough exploratory data analysis of the measurements before the application of the long-term correction method. The choice of the appropriate MCP method(s)

may then depend on:

– Data availability and correlation of measurement and reference dataset.

– The deviation of the wind climate in measurement period from the climate mean.

– The general wind climate.

– The complexity of the terrain at the individual site.

**4  Discussion & Conclusions**

This study contributes to an improved assessment and a reduction of uncertainty due to inter-annual variability in the long-term referencing process. It quantifies the impact of inter-annual - i.e. the year-to-year wind variability - on hindcast wind speed predictions using seven different correlation methods of varying complexity within the measure-correlate-predict (MCP) approach. This analysis benefits from quality-controlled, multi-year tall tower measurement data from twelve wind energy-

relevant sites at the height of modern wind turbines. While these data are rare, they offer valuable insights into wind energy related questions. The study provides an overview of the uncertainty related to the inter-annual wind variability across different site complexities and emphasizes the substantial impact that a single measured wind year - as commonly done in the wind



energy industry - can have on long-term referencing. Specifically, the following recommendations and conclusions can be made:

– **Quantification of inter-annual variability:** Inter-annual variability has been investigated in several studies. In the overview of Lee and Fields (2021), the uncertainties related to inter-annual variability of wind are indicated to be up to 10 %, with most studies reporting uncertainties around 5 %. This is generally consistent with our findings, which indicate an average of 6.5 %. Notably, the obtained variability across different sites and correlation methods spans from 1 % to 14 %, suggesting that relying solely on such an average estimate may be inadequate. Therefore, it is essential to
investigate factors that can contribute to narrowing this range.

– **Time range dependency:** Excluding years with a wind climate deviating strongly from the climate mean reduces the average by approx. two percentage points to 4.2 %. Furthermore, based on a smaller subset of tall tower measurements we investigated the impact of using individual yearly measurements from a dataset of 15 years of measurements vs. data from seven years of continuous measurements. These results indicated that the full variability was not covered in
the shorter dataset even after the application of different methods in the MCP correction process. Other - purely model-based - studies indicate significant multidecadal variability Wohland et al. (2019).

– **Database:** Discrepancies between model based long-term reference and short-term measurement data can vary depending on factors such as horizontal and vertical resolution or model physics but also the measurement data quality at each site. Data gaps can have an impact on the results with varying magnitude depending on the methodology. Thus, this study
demonstrates the importance of long-term - i.e. multi-year - well documented observational data from tall meteorological towers across various regions (terrain complexity, geographic location). In their comprehensive review (Carta et al., 2013) summarize that the database has a higher influence on uncertainties than the choice of the correlation method in the MCP process. Our study aligns with these findings; however, it also demonstrates that employing the appropriate correlation method can be beneficial to reduce uncertainties in the long-term referencing process associated with inter-annual
variability.

– **Correlation method:** We show that the selection of the correlation method in the MCP process can have an influence on the variability of the long-term corrected wind speed. This is particularly true when the atmospheric conditions in the short-term measurement data are not representative of the wind climate, for example, due to extraordinary weather. Accordingly, we recommend not to rely just on one method, but rather test different methods and pay attention to the
variability.

The sector-wise linear regression method exhibits high sensitivity to the input data. Insufficient data per sector can result in inadequate correction functions. To address this issue, a dynamic sector detection approach as proposed by (Riedel et al., 2001) and (King and Hurley, 2005) can be used. We did not investigate such corrections to enable a comparison of a simple implementation without any site specific or other manual adaptations.



On average, the hindcasts based on the more advanced ML methods Random Forest (Grömping, 2009) and AdaBoost (Drucker, 1997) as well as the SpeedSort (King and Hurley, 2005) method are less affected by the inter-annual variability in the measurement data. Reductions appear to occur in heterogeneous and complex terrain, while no reduction is noted in simple and very complex terrain. To further substantiate this, a larger number of sites per complexity category would be beneficial. Further, in this study, the same set of features was used within the advanced ML approaches for all sites

enabling a comparison of the correlation methods. Site specific calibrations of ML methods could potentially contribute to further improving the individual results but reduce the comparability.

    In summary, this study offers supportive insights into uncertainties in long-term referencing due to inter-annual variability which is particularly relevant in the planning and operation phase of wind farms. Besides a thorough investigation of long-term hub height wind measurement data only a small number of sites with suitable multi-year data could be investigated. In case

further high-quality measurement data should become available, future studies could extend to further areas of various terrain complexity. The impact of climate change was not considered in this study, further studies could make use of climate model data in the MCP process to cover future wind variability.

*Data availability.* The ERA5 model data are made publicly available via the Copernicus Climate Change Service - Climate Data Store (https://climate.copernicus.eu/). The tall mast data Megler, Goodnoe Hills, Chinook, Butler Grade, Boulder - NWTC M2 (Jager and Andreas,

1996) and Park Falls (Davis et al., 2003) were obtained from the Tall Tower Dataset (Ramon et al., 2020) and are publicly available via https://talltowers.bsc.es/. Data from the Cabauw tall met mast are freely available at https://dataplatform.knmi.nl/dataset/?res_format=NetCDF&tags=Cabauw. The tall mast data with the locations Hamburg, Karlsruhe (Kohler et al., 2018) and Cabauw can be requested from the individual data owners.

*Author contributions.* JB: Data analysis, writing - original draft, review and editing. SaS: Assisting with MCP implementation, discussion,

writing and review. KA: Discussion, writing and review. MD: Conceptualization, discussion, writing, review and editing.

*Competing interests.* The authors declare that there are no competing interests.

*Acknowledgements.* The results presented in this paper were derived in the framework of the KliWiSt (grant no. 03EE3041A) project. The KliWiSt project was funded by the German Federal Ministry for Economic Affairs and Climate Action (Bundesministerium für Wirtschaft und Klimaschutz–BMWK) due to a decision of the German Bundestag. K.A. acknowledges funding from the Ministry of Science and

Culture of Lower Saxony through the "Zukunftskonzept Windenergieforschung". We thank the ECMWF for providing the ERA5 reanalyses data and making them freely accessible. We would also like to thank everyone who contributed to the Tall Tower Dataset (Ramon et al., 2020) for providing access to the met mast data. Additionally, we thank the following individual organizations for granting access to the





tall met mast data: The Royal Netherlands Meteorological Institute (KNMI) for the Cabauw data by the Cabauw Experimental Site for Atmospheric Research (CESAR), the Deutscher Wetterdienst – Meteorological Observatory Lindenberg - Richard-Aßmann-Observatory

(DWD/MOL-RAO) for the data of the Boundary Layer Field Site (GM) Falkenberg, the Institute for Meteorology and Climate Research (IMK) - Karlsruhe Institute of Technology (KIT) for the data of the met mast at KIT and the Meteorological Institute of the University of Hamburg (UHH-MI) for the Hamburg Weather Mast data. We want to thank Julie K. Lundquist for her initial discussion during the early phase of the project.



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
