# Peer review of "Evaluating the Impact of Inter-Annual Variability on Long-Term Wind Speed Predictions"

_Wind Energy Science, 2025_

## Author Comment (AC1)

**Reply to Referee 1 by Borowski et al..**

We thank the referee for the very helpful comments that have significantly contributed to improving the manuscript and have addressed each comment below. Our answers are given in "italics". The line numbers given refer to the initial version of the manuscript.

**General comments:**

This paper makes a good contribution to the literature on evaluating long-term variability in the wind resource, an ongoing challenge for wind energy science. The authors make good use of a set of long-term (by wind energy standards) tall-tower data sets to compare several methods to estimate multidecadal wind speed variability from much-less-than-multidecadal in-situ observations. They apply several MCP methods and three widely used ML methods to data from sites in different geographical and climatological settings. The ideal outcome would be to determine the best method(s) to use given a particular geographic or climatological setting, but given the complexity of the wind resource, the results are a little more nuanced than that, and understandably so. Nonetheless, the authors describe how the methods compare to each other across these settings and are able to offer some tentative conclusions and recommendations for estimating long-term variability from short-term records.

⇒ Thanks for your comment and your positive view on our manuscript. We agree that we would have been happy to determine more generalized recommendations, which is difficult - as you noted - due to the complexity of the wind resource at the various sites.

**Specific comments:**

- (1) Lines 100-105: The ERA5 data set begins in 1940 but your analysis begins in 1950. Given your goal of characterizing long-term variability, why exclude this additional 10 years of reanalysis data?
- ⇒ Thank you for the comment. The backward extension of ERA5 has been publicly available since 2023 (Soci et al., 2024). The original design of the study began before that. Thus, we started with including data from 1950 onwards. The quality of reanalyses has steadily improved over time as more and more observational data has been integrated over the decades. Studies indicated that especially data for the decade prior to 1950 are much poorer in particular in Europe resulting in lower quality reanalyses (Soci et al., 2024). Thus, we decided to not extend our analysis backwards when the data became available.
- (2) Table 2: Are all the sites freestanding met towers or are they towers in the vicinity of a wind turbine, which might possibly be affected by wakes from some wind directions?
- ⇒ Thanks for this comment, we checked every location carefully. Indeed, there are some sites with wind farms in the immediate surrounding, e.g. Goodnoe Hills and Butler Grade. However, turbines are not in dominating wind direction sectors and should thus not have a relevant impact on our analyses. A respective sentence has been added to the Section 2.1.1 (line 88): "Some masts, e.g. Butler Grade and Goodnoe Hills have wind turbines in the surrounding but not from relevant wind direction sectors. So, the majority of sites are not impacted by wakes."
- (3) Line 210: The headings in Table 4 list 2010-2016 and 2012-2016, but the text says the MER is for 1950-2020. I suggest adding that bit of info to the table heading, just as a reminder to readers.
- $\Rightarrow$  Thank you for this comment, we revised the caption of Table 4 and changed it to: "Overview of the mean max. hindcast ensemble range (MER [ms-1]) for the time period from 1950 to 2020 and its normalization (MNER [%]) by the mean measured wind speed. The ensemble is based on the individual years within the time intervals 2010 to 2016 and 2012 to 2016."

- (4) Lines 220-227: I was confused at this point about why you would want to reduce the influence of interannual variability when creating a long-term reference data set. You talk about this a bit later in the paper, but maybe add a note here that you'll come back to this in section 3.4.1? In general, I think it might be useful to say a little more in the paper about the importance of long-term "data" such as reanalyses, which in theory can include ENSO and other climate patterns that influence long-term variability at a site.
- ⇒ Thanks for the comment. To give the reader a better orientation, we have included a note (line 224) directing readers to the detailed discussion in the respective section: "The impact of the years 2010 and 2011 is further analyzed and discussed in greater detail in Section 3.3.1 using various MCP approaches."

Further, we added some more information about the importance of long-term data to the manuscript in the introduction, data and discussion section:

"Introduction (line 49): Long-term datasets, such as reanalysis data that integrate model outputs with actual measurements, can capture and reflect these effects. Consequently, long-term data are crucial for analyzing historical trends, variability, and anomalies in climate patterns, making them essential for assessing long-term wind conditions.

"Section 2.1.2 (line 94): Current generation of reanalysis datasets provide data over periods of more than 50 years and therefore multi-decadal records of atmospheric conditions, enabling the study of large-scale climate patterns and their impact on (regional) wind variability.

"Discussion (line 364): Long-term data from reanalyses enable the analysis of historical climate parameters and patterns, particularly their variations and temporal developments, and are therefore essential for estimating wind resources. Although reanalyses largely depict fluctuations and patterns in a uniform manner, differences in model physics and resolutions lead to discrepancies. Expanding the study to include various reanalyses could help quantifying the sensitivity to long-term data and reducing related uncertainties.

- (5) Line 240: You note here, and in several other places in the paper, that the MNER doesn't seem to depend much on terrain complexity. Do you have any hypotheses as to why? I encourage you to include a bit more discussion of this even if only possible hypotheses in the Discussion and Conclusions section (e.g., lines 380-383). Perhaps that discussion can also touch on your comments about sensitivity to sample size and data gaps (lines 259-263).
- ⇒ Thank you for the comment. We have revised the manuscript and included additional sentences in the discussion section (line 354): "A generalizable dependency of these values on terrain complexity could not be identified. This could be an indication that the variability is more dependent on other characteristics of the local wind climate such as the homogeneity of the wind direction distribution, the land-use in the surroundings or the quality of the measurement data (see "Database" below)."

**Technical comments:**

- (6) Figure 9: The figure caption should say 2012-2016, not 2010-2016.
- $\Rightarrow$  Thanks for the comment. Changed accordingly.
- (7) Figure 10: The figure caption shows 2010-2016 (MNER2010-2016 %) twice; the second one should be 2012-2016.
- ⇒ Thanks for the comment. Changed accordingly.

**References**

Soci, C., Hersbach, H., Simmons, A., Poli, P., Bell, B., Berrisford, P., et al.: The ERA5 global reanalysis from 1940 to 2022, Quarterly Journal of the Royal Meteorological Society, https://doi.org/10.1002/qj.4803, 2024.

---

## Author Comment (AC2)

**Reply to Referee 2 by Borowski et al..**

We thank the referee for the very helpful comments that have significantly contributed to improving the manuscript and have addressed each one accordingly below. Our answers are given in "italics". The line numbers given refer to the initial version of the manuscript.

Borowski et al. provide an interesting assessment of the impact of inter-annual variability on long-term wind speed estimates. The topic is a valuable one and the authors did a nice job with their research design and analysis. I especially appreciate the helpful tables describing the observational sites and the variety of ML models considered. The manuscript is a bit challenging to follow at times, with references to analyses discussed much later in the text than the reader's current location, but the findings are nicely summarized in the discussion section.

⇒ Thank you for your thoughtful feedback and positive comments on our research design and analysis. We are pleased that the discussion section provides a clear and effective summary of our findings, despite difficulties in following at times. The other referee had a similar comment. Thus, we added a reference to a later section at one point.

**General comments:**

**Comment -** There are a number of extremely short paragraphs (two sentences, sometimes only one) in the text that should either be elaborated upon, combined with other relevant text, or removed if they are found to provide limited context or support for the analysis.

- ⇒ Thank you for the comment. We agree and removed unnecessary line breaks:
  - Line break 20-21 removed.
  - Line break 22-23 removed.
  - Line break 62-63 removed.
  - Line break 77-78 removed.
  - Line break 82-83 removed.
  - Line break 88-89 removed.
  - Line break 102-103 removed.
  - Line break 156-157 removed.
  - Line break 160-161 removed.
  - Line break 193-194 removed.
  - Line break 197-198 removed.
  - Line break 291-292 removed.
  - Line break 296-297 removed.
  - Line break 303-304 removed.

**Comment:** This paper's value to the wind energy community would benefit significantly if the results were translated into an energy parameter in the discussion. In particular, I could see great value by running the various wind speed time series through a reference turbine power curve and then speaking to the findings in terms of capacity factor ranges.

⇒ Thank you for this valuable comment. We agree that translating the results into an energy parameter could enhance the paper's relevance to the wind energy community. However, using a single reference

turbine power curve for all sites might introduce a bias, as turbines are typically selected based on site-specific wind conditions, which vary significantly across the investigated sites. To address this and providing meaningful energy-related insights, we additionally calculated the "mean normalize ensemble range" (MNER) for wind power density (WPD) at each site, as well as, the factor between MNER of wind power density and wind speed and included both as Figure and corresponding text in Section 3.3 (initial manuscript). This approach allows us to express the results in terms of wind energy potential while avoiding using a turbine power curve for all locations. The following additions to the manuscript were made:

Method Section 2.2.2 (line 167): "The wind power density (WPD) is calculated by WPD =  $0.5\rho v^3$ , where v represents wind speed. The air density  $\rho$  is determined by  $\rho = p_0(RT)^{-1}$ , with  $p_0$  being the surface pressure, R as the specific gas constant of dry air ( $R = 287.058 \ J(kg \cdot K)^{-1}$ ), and T is the the surface air temperature at 2 meters. The WPD-MNER is calculated similarly to the MNER of wind speed (as described in the previous paragraph). The factor between WPD-MNER and MNER of wind speed is calculated by the quotient: WPD-MNER / MNER of the wind speed."

Figure 9: Left: Comparison of the mean max. normalized ensemble range based on the multi-year overlap time period 2010–2016 for wind power density (WPD-MNER2010–2016 [%], illustrated by the color scale) as obtained by different methods (indicated below) and for varying sites (indicated on the left). The sites are sorted by the terrain complexity from lowest to highest (black arrow); terrain complexity categories (simple, heterogeneous, (very) complex) are separated by dashed horizontal black lines. Right: Figure setup like left, but for the factor between WPD-MNER2010–2016 and MNER2010–2016 for wind speed.

Result Section 3.3 (line 274): "The relationship between wind power density and wind speed is cubic and therefore, uncertainty in estimating wind speed significantly impacts the uncertainty of wind power density and annual energy potential. The translation of wind speed uncertainty to annual energy potential uncertainty is not consistently defined. Holtslag (2013) state that 1% uncertainty in wind speed is related to 1.8% in annual energy potential. EMD (2025) give the rough orientation that in regions with low wind speeds (6-7 m s-1), wind speed uncertainty should be tripled for annual wind potential uncertainty while it is only doubled for higher wind speeds (about 8 m s-1) and only 1.5 times at regions with even higher wind speeds (about 9 m s-1). To connect the MNER of wind speed (Fig. 8) to a more comprehensive energy parameter, the MNER for wind power density (Fig. 9, left) is calculated.

Additionally, the factor (Fig. 9, right) between the MNER of wind speed and wind power density is determined. The results indicate that, on average, a factor of approximately three (2.6) is achieved. Further, the factor between MNER of wind speed and wind power density does not increase with complexity. But, the studied sites have averaged wind speeds ranging from approximately  $5\,\mathrm{m\,s^{-1}}$  to  $7\,\mathrm{m\,s^{-1}}$  (Table 1), which suggests that the roughly tripled uncertainty aligns with the results from EMD (2025)."

Section 3.3 (line 276): "The observed tripled uncertainty in WPD-MNER underscores the importance of accurately estimating wind speed and reducing wind speed uncertainties."

Discussion (line 356): "Translation into energy parameter: The analysis of the relationship between wind power density and wind speed reveals a cubic dependency, indicating that even minor uncertainties in wind speed can lead to significant variations in annual energy potential. Transferring the results from wind speed uncertainty to wind power density, an average factor of 2.6 was found; but, it could be individually higher or lower. Our study, which encompasses sites with wind speeds between  $5\,\mathrm{m\,s^{-1}}$  and  $7\,\mathrm{m\,s^{-1}}$ , aligns with the EMD (2025) assertion of approximately tripled uncertainty under these conditions. This significant factor highlights the critical need for accurate uncertainty estimations in wind speed predictions, with the goal of enhancing the reliability of energy yield prediction in the wind energy sector."

Discussion (line 384): "When translating wind speed uncertainties to wind power density uncertainty, SpeedSort appears to be the most affected, with a factor of 3.7, while Random Forest and AdaBoost exhibit similar performance to the classic methods, with a factor of approximately 2.5."

**Specific comments:**

**Comment - Line 29:** I recommend removing the word "hindcast", as this method is often applied forward in time as well, as reference datasets increase their temporal coverage.

 $\Rightarrow$  Thank you for the comment. We have revised the sentence and removed the term "hindcast" in this case. Please see our response to "Comment - Section 2.2.2" regarding why this change was not applied to other instances in the manuscript.

Comment - Line 40: Diurnal uncertainty is another important component that deserves mention.

- ⇒ Thanks, we revised the sentence to mention the diurnal uncertainty as well (line 41):
  "..., wind variability (diurnal, inter-annual and inter-monthly, as well as future projections), ..."
- Comment Line 50: Concerning the more than 20 methods Lee et al. (2018) compared, can you follow up with a quick mention of which method(s) was found to be most advantageous and why?
- $\Rightarrow$  Thank you for the comment. As suggested, we added a few sentences about Lee et al. (2018)'s findings to the introduction (line 51):
- "They recommend using the robust and resilient coefficient of variation (RCoV) to quantify the fluctuations in wind resources and energy production. Since RCoV is a normalized spread metric and compared to other metrics tested more effectively captures the relationship between large wind speed fluctuations in a wind farm and the resulting variability in wind energy production, it is an advantageous spread metric compared to other metrics. However, ..."

Comment - Line 90: When aggregating to hourly intervals, are you selecting the top of the hour measurement (which would best align with ERA5's instantaneous hourly output) or converting to hourly averages?

 $\Rightarrow$  Thanks for the question. Since 10-minute averages are available (for most of the sites) and the measurement at the top of the hour corresponds to either the 10-minute average before or after the

hour (depending on the site), it likewise does not reflect the instantaneous ERA5 value. Averaging the two 10-minute intervals before and after the full hour could be used to approximate the instantaneous ERA5 value. However, the resulting value differed only marginally from the hourly mean. So, we decided to take hourly averages of the measurements. In addition, due to the coarse spatial resolution of the ERA5 reanalysis the fluctuations on hourly level typically also represent the conditions better than in 10-min intervals that still contain fluctuations that are not present in the reanalysis dataset.

**Comment - Line 94:** "Modern reanalysis..." – you may wish to update the grammar and intention of this sentence for clarity.

⇒ Thank you for the comment. We have revised the sentence and have changed the wording from "Modern reanalysis" to "Current generation of reanalysis".

Comment - Line 98: This sentence may prove controversial in its current form. Many wind energy researchers feel that a 1-hour temporal resolution is not high and they do not attribute their use of ERA5 to that or the relatively coarse spatial resolution. Rather, many studies show that ERA5 exhibits good correlation with wind observations. You may wish to focus on this aspect, as it ties nicely to MCP.

⇒ Thank you for this comment. We have revised the text section as follows:

"While mesoscale dowscaling approaches might provide even more accurate average wind speeds at sites, several studies have pointed out the suitability of ERA5 for wind energy applications (Olauson, 2018; Hahmann et al., 2020; Dörenkämper et al., 2020; Gottschall and Dörenkämper, 2021) due to its relatively high spatial resolution of ~0.28°(31 km) and - compared to other reanalyses - high resolution in time of one hour. In particular, the correlation between measurements and ERA5 has been proven to be high, making it in particular suitable for long-term referencing applications (Meyer and Gottschall, 2022; Gottschall and Dörenkämper, 2021)."

**Comment - Line 103:** Do you have confirmation that the nearest grid points are on land and not over water, where the dynamics can be very different? Some of your observations are quite close to the coast.

 $\Rightarrow$  Thank you for your comment. Indeed, as you noted, the nearest model grid point for the site Megler is situated over water, while all other sites were not affected. To address this, we recalculated using the next onshore grid point from the ERA5 reanalysis. We have revised the manuscript accordingly and added a sentence to Section 2.1.2:

"An exception is made for the Megler site, where the nearest grid point is over the sea, and the dynamics can differ significantly. Therefore, the next grid point that has land surface properties in the reanalysis was selected instead."

During the recalculation, changes in the python-based implementations of the machine learning methods of the selected libraries affected values at other sites for RForest, but especially for AdaBoost. The uncertainty related to these implementation changes varies by site, averaging between 0.0 and 0.3 for RForest, and reaching up to 0.6 for AdaBoost. For other methods, only minor rounding errors were observed. We carefully revised the manuscript and updated the values. The overall conclusions of the study of course remains unchanged, and therefore, no modifications were made to the context of the manuscript. To address the uncertainty related to the change of implementation of the machine learning methods, we added another sentence to the discussion (line 384) and revised the surrounding:

"Further, changes in the implementation of machine learning methods can influence the outcome. The impact of the changes in the implementation is highly dependent on the specific site, resulting here in average values ranging from 0 to 0.3 for RForest and reaching up to 0.6 for AdaBoost."

Comment - Line 105: Why are you only focusing on 1950 to 2020 when ERA5 has 1940 to present available?

⇒ Thank you for the comment. The backward extension of ERA5 has been publicly available since 2023

(Soci et al., 2024). The original design of the study began before that and we started with including data from 1950 onwards. The quality of reanalyses has steadily improved over time as more and more observational data has been integrated over the decades. Studies showed that especially data for the decade prior to 1950 are much poorer especially in Europe, resulting in lower quality reanalyses (Soci et al., 2024). Thus, we decided to not extend our analysis backwards when the data became available.

**Comment - Line 148:** This sentence is a bit confusing and implies that temperature and pressure are decomposed into sine and cosine, which Table 3 contradicts.

 $\Rightarrow$  Thanks, it was not our intention to express this. We revised the text to provide clarity: "The wind speed related and atmospheric state descriptive variables wind direction, temperature, pressure, time of day, and seasonality are used as features for the advanced ML approaches. The variables with circular nature (wind direction, time of day, months) are decomposed into sine and cosine components (Table 3)."

**Comment - Section 2.2.2:** Again, I encourage use of a different term than "hindcast" as Figure 3 shows that you look forward in time from the measurements as well as behind. Perhaps "Long-term ensemble range"?

⇒ Thank you for the comment. Indeed, from the perspective of the used measurements, the long-term correction is also looking into the future; viewed from today, the entire corrected times series lies in the past. Because of this and to maintain consistency throughout the manuscript, we have retained our wording. But to ensure clarity, we have added a clarifying note within the method section (line 156). "Note, that the long-term correction covers periods both before and after the measurement period, we use the term "hindcast" because the entire corrected time series lies in the past from today's perspective."

**Comment - Section 3.1: Might align better in Section 2.**

⇒ Thanks for this comment, we have moved it to the recommended section and revised the surrounding paragraphs and headings. The previous section 3.1 has been renamed to "Wind climatology at selected sites: A comparison with ERA5," while section 2.1 is now titled "Selection of met mast and reanalysis data" for clarity regarding the content of each section.

Comment - Line 205: Can you please be more specific than "see below"? Perhaps guide the reader to a specific section, figure, or table.

⇒ Thanks for the hint, we changed "see below" to "see Section 3.3.1".

**References**

EMD: WindPRO 4.2, User Guide, EMD International A/S, https://help.emd.dk/knowledgebase/content/windPRO4.2/c3-UK\_windPRO4.2\_ENERGY.pdf,Page 41, Figure 40, 2025.

Gottschall, J. and Dörenkämper, M.: Understanding and mitigating the impact of data gaps on offshore wind resource estimates, Wind Energ. Sci., 6, 505–520, https://doi.org/10.5194/wes-6-505-2021, 2021.

Holtslag, E.: Improved Bankability: The Ecofys position on LiDAR use,https://www.nrgsystems.com/assets/resources/Ecofys-2013-position-paper-on-lidar-use-Whitepapers.pdf, Utrecht, the Netherlands, 2013.

Meyer, P. J. and Gottschall, J.: How do NEWA and ERA5 compare for assessing offshore wind resources and wind farm siting conditions?, Journal of Physics: Conference Series, 2151, 012 009, https://doi.org/10.1088/1742-6596/2151/1/012009, 2022.

Soci, C., Hersbach, H., Simmons, A., Poli, P., Bell, B., Berrisford, P., et al.: The ERA5 global reanalysis from 1940 to 2022, Quarterly Journal of the Royal Meteorological Society, https://doi.org/10.1002/qj.4803, 2021.